# The Phosphorus Transport in Groundwater from Phosphogypsum-Based Cemented Paste Backfill in a Phosphate Mine: A Numerical Study

**DOI:** 10.3390/ijerph192214957

**Published:** 2022-11-13

**Authors:** Qiusong Chen, Huibo Zhou, Yikai Liu, Daolin Wang

**Affiliations:** 1School of Resources and Safety Engineering, Central South University, Changsha 410083, China; 2Sinosteel Maanshan General Institute of Mining Research Co., Ltd., Ma’anshan 243000, China; 3Department of Geosciences, University of Padova, 35131 Padova, Italy

**Keywords:** cemented paste backfilling, green mine, phosphogypsum, leaching, phosphorus transport

## Abstract

Stacked phosphogypsum (PG) can not only cause a waste of resources but also has a serious negative impact on the surface environment. Phosphogypsum backfilling (PGB) in the underground goaf is a useful approach to effectively address the PG environmental problems. However, the effects of this approach on the groundwater environment have not been studied. Therefore, the present study aims to assess the spatiotemporal evolution mechanism of total phosphorus (TP) in groundwater to solve the diffusion regular pattern of TP in PGB bodies, as well as to manage and mitigate the impacts of TP on the groundwater system. In this study, leaching toxicity experiments and a numerical groundwater simulation software (GMS10.4) were combined to develop a three-dimensional conceptual model for predicting the groundwater flow and contaminant transport under steady-state conditions in a phosphorus mine in Anhui. The results showed a lower TP concentration than the TP standard concentration (0.2 mg/L) at a source concentration of 0.59 mg/L. However, groundwater TP source concentrations of 1.88 and 2.46 mg/L in the study area were found to exceed the standard concentration for a certain time and areas. In addition, the transport and dispersion of TP are influenced not only by the groundwater flow field, drainage ditches, rivers, and wells but also by the adsorption and attenuation effects of the soil that occur during the transport process, affecting the dispersion distance and distribution of groundwater TP concentrations. The results of the present study can promote the development of groundwater-friendly PGB technology, providing a great significance to the construction of green mines and the promotion of ecological civilization.

## 1. Introduction

Phosphogypsum (PG) is a solid waste generated during the manufacturing of phosphoric acid [1]. This solid waste contains large amounts of phosphorus pentoxide (P_2_O_5_), fluorine (F), organic matter, potentially radioactive elements, and other impurities [2]. Indeed, about five tons of PG are generated from every ton of phosphoric acid which is produced [3]. According to previous reports, China’s PG production in 2019 was estimated at 75 million tons, representing about 25% of the global PG production (300 million tons) [4,5]. However, the utilization rate range of PG in China is only 30–40%. Since most of the PG is stockpiled, it not only causes a waste of resources but also a serious pollution of the surrounding environment [6,7,8,9]. Therefore, considering PG as a mine backfilling material can not only improve the resource recovery rate and address the problem of PG accumulation but also ensures the sustainable development of mines [10,11].

Although the treatment of a PG backfill involves, essentially, the dumping of PG into the underground goaf [12], the impact of toxic and hazardous substances precipitated from the backfilling body on the underground environment limits the application of phosphogypsum backfilling (PGB) technology. Yin et al. [13] found that the PG generated during phosphoric acid production contains high levels of arsenic (As), resulting in adverse effects on the surrounding environment, as well as on human health when PG is stacked or handled. Chen et al. [14,15,16,17,18] conducted toxic leaching tests on PGB specimens and observed precipitations of small amounts of phosphorus, fluorine, and other elements from PG even under the action of cement curing. Shi et al. [19] observed a large amount of phosphate (P) in the leached PG, resulting in negative impacts on the surrounding environment, even though the phosphorus concentration was reduced, using modified quartz sand combined with a backfilling slurry. Therefore, although the pollution of the surrounding environment can be slightly reduced following the backfilling of the underground goaf with PG, the total phosphorus (TP) and other pollutant ions still cause serious impacts on the soil and groundwater [20,21,22,23,24,25,26,27,28,29]. The slow-release characteristics of phosphorus, as a representative pollutant of the PG load, were assessed previously by several researchers, while the research results of the diffusion pattern of pollutants in PG loads to groundwater are limited. Indeed, China’s groundwater management regulations (100022-2022-00063) were adopted by the State Council in September 2021 to strengthen groundwater management, as well as to prevent and control groundwater overdraft and pollution, ensuring a good quality and the sustainable use of groundwater and promoting ecological civilization construction. Therefore, assessing the transport and dispersion patterns of TP in the groundwater environment can not only provides effective guidance for the prevention and control of TP pollution in groundwater but it also plays an important role in protecting the groundwater environment [30].

The diffusion of pollutants in the groundwater environment has a significant impact on the ecological environment and can even directly affect human life and health. Indeed, researchers in all environmental fields have devoted great attention to this issue and used various methods to assess the diffusion and transport of pollutants in groundwater to protect the groundwater environment and thus human life and health [25,26,27,30,31,32,33,34,35]. Zhang et al. [21] studied the spatiotemporal transport and geochemical evolution of heavy metal elements and assessed the human health risk in the contaminated areas of groundwater with heavy metals to provide further insights into the prevention of specific potential negative impacts on human health. Mishra et al. [36] evaluated the impact of waste leachate on the surrounding groundwater quality at Ramna village landfill in Varanasi, India using a physicochemical analysis of waste and the leachate pollution index (LPI). They found that it was not safe for drinking purposes as most of the physio-chemical parameter values exceed the permissible limit of the drinking water standard. On the other hand, Zeng et al. [23] simulated the transport of pollutants during groundwater drainage using the finite element subsurface flow (FEFLOW) model and determined the factors influencing the transport of pollutants, proposing an economical and reliable long-term monitoring program for controlling the groundwater pollution in mining areas. Whereas Sathe et al. [37] developed a conceptual three-dimensional transient prediction model for the groundwater flow and contaminant transport in two As-contaminated areas with a different topography using numerical groundwater simulation software (GMS). The simulation results showed that the distribution of As concentrations was directly controlled by the complex hydrostratigraphy and surface water quality and indirectly controlled by the variation in the meteorological conditions, while the downward movement of As makes the deep aquifer unsuitable for drinking and irrigation purposes. The results of these scholars provided important insights into the transport and dispersion of TP in groundwater and demonstrated the effectiveness of mathematical modeling methods in groundwater contaminant dispersion studies. However, the diffusion pattern of total phosphorus, based on the toxicity leaching characteristics of PGB bodies, has not been assessed, and there is a lack of the related mathematical models with a reference value.

This study aims to explore the spatiotemporal evolution mechanism of PG-derived TP in the groundwater environment around a phosphate mine in Anhui Province under different leaching concentrations. The results of the toxicity leaching tests were combined with numerical groundwater simulations to construct a groundwater TP transport and dispersion model. The spatiotemporal transport and dispersion equations of TP were established by fitting and solving the TP transport and dispersion equations results using the 1stOpt software package.

## 2. Materials and Methods

### 2.1. Numerical Simulation

#### 2.1.1. Study Area and Stratigraphy

This numerical simulation is based on the geological background of a phosphate mine in Anhui Province, China. Since the transport and dispersion of PGB in the groundwater environment is studied, the study area is divided into eastern and western parts according to the geological conditions around the phosphate mine for the simulation study separately, as shown in Figure 1. The mine area is mostly low-altitude hilly terrain with a low topography in the south and a high topography in the north, and the topography decreases from the northwest to southeast, with an undulating topography in the north and a flat topography in the south. The mine area has a subtropical continental climate: it is warm, humid, and rainy, with an average annual temperature of 16.1 °C, an extreme maximum temperature of 41.3 °C, and an extreme minimum temperature of −12.3 °C. The dominant wind direction is southwest. The average annual rainfall is 1174.2 mm, and the relative humidity is 77~80%. The average annual snowfall days are 11.9 days, and the frost-free period is 246~255 d. The Brick Bridge Formation is the main rock layer within the mine area, followed by the Yangwan Formation red layer and the Shuangmiao Formation red layer. The groundwater is mainly fissure water, there are 4 water-bearing layers, and 1 layer of the water barrier. The fissure development of the aquifer is low and contains weak fissure water. There is no large surface water body in the vicinity of the deposit, and the rock layer is weakly permeable and has a limited recharge. Although the hydrostatic pressure is high, the hydrostatic reservoir is the main source of water, which can be easily drained. The ore body and its roof rock do not contain water, and most of the water between the ore body and the secondary quartzite aquifer is separated by kaolinization and mudification. Therefore, the hydrogeological conditions of the deposit are of a medium type.

#### 2.1.2. Hydro-Geological Conceptual Model Development

The study area was gridded into a total of 44908 horizontal grids and a discrete vertical grid of 3 layers, with a dimension of 4356 m (x) × 2058 m (y) × 10 m (z). The limestone in the study area was defined in the north as a zero-flow boundary, while the average water levels of the river in the southeastern and southwestern parts were represented by a fixed-head boundary. The two main sedimentary layers, namely the upper and lower layers, were considered to be unconstrained and constrained, respectively, in the model. In addition, the riverbed in the study area was defined as a drainage ditch since it can be occasionally filled with groundwater. Two production wells in the study area were included with constant flows of 50 and 300 m^3^/d.

#### 2.1.3. Parameter Settings Related to TP Diffusion in MT3DMS

Since the calculation of the flow solution using the MODFLOW module showed a steady state in this study, the desired sequence of the stress cycles and time steps were simulated. In addition, based on the MODFLOW simulation results for the study area, it was concluded that the water levels of the two rivers and groundwater recharge in the simulated area have important influences on the groundwater levels. Moreover, the drainage ditch and flow boundary in the western and southern parts of the study area were important sink parameters for the groundwater flow simulation. On the other hand, since the TP in the PGB bodies can be released at a constant rate, the diffuse transport of TP in groundwater in the study area is simulated for 100 years. The MT3DMS module was used to calculate the appropriate transport time step by setting the transport step to zero. Indeed, the solute transport in the pocket gas zone is extremely complex, especially in unsaturated water conditions, and has been understudied worldwide. Therefore, only the adsorption of TP and the attenuation process were considered in the simulation of the TP transport through the soil in the inclusion zone under unsaturated water conditions. Adsorption can delay the flux movement and decay, reducing the concentration of the pollutant by the biodegradation process. Indeed, the ability of soils to adsorb P varies greatly among different soil types since it is closely related to the properties of the soil (e.g., the clay content, clay mineral type, pH, and the organic matter content).

Flora Amarh et al. [38] collected four soil samples from the eastern region of Ghana for P sorption measurements by equilibrating air-dried soils (<2 mm) in 0.2 M of CaCl_2_ solutions containing various concentrations of P. The observed P sorption data in the soils were fitted to Langmuir, Freundlich, and Temkin isotherm equations. According to the obtained results, all the sorption isotherm equations described well the P sorption. This study area is dominated by sediments and limestone bedrock, whose soil components have a good affinity for P. Therefore, in this study, the adsorption of TP during the transport process in the filled body was investigated using the Freundlich isotherm, according to the following equation:(1)$$W/{m=kC}^{1/n}$$
(2)logW/m=logk+logC/n
where *W* is the mass of the solute adsorbed on the adsorbent mass m. *C* is the equilibrium concentration of the solution when *k* and *n* are the constants. From the equation, the parameters of importance are the sorption capacity (*k*) and P sorption energy (*n*).

Meanwhile, the adsorption and decay rates of TP in the groundwater environment use first-order irreversible kinetic reactions. Serguei A. Bobrovnik [39] developed a method for solving the rate constants and total end products in the first-order irreversible kinetic rate reaction equations based on the analytical solution of the transcendental system of the equations for the reaction kinetics in special cases. Suppose the initial species is A and the final to product is B. According to the Guggenheim method, the concentration of species B is measured at t_i_ and t_n_ moments and substituted to calculate the kinetic rate reaction constant. In the case of uncertainty of the B concentration, the rate constant k is determined. The derivation is shown in Figure 2.

### 2.2. Test Material

The PG was collected from phosphate mines in Tongling, Anhui Province, China. The collected 10 kg PG sample was hermetically encapsulated in plastic barrels to prevent changes in their properties. The PG was first over-dried at 58 ± 2 °C for 24 h, then crushed evenly using a crushing machine, sealed, and stored under airtight conditions [38,40].

The physicochemical properties of PG have an important influence on the results of the properties of the backfilling bodies and the leaching toxicity tests. First, XRD and XRF experiments were used to detect the basic composition of PG, and also to provide a reference basis for the leaching toxicity experiments. By analyzing the particle size composition of PG, the particle size distribution of PG is determined, which provides a basis for the proportioning of backfilling experiments. Before conducting the tests, the particle size composition of PG was first determined using a laser diffraction particle size analyzer (Mastersizer 3000), while the mineralogical composition of PG was determined using an X-ray diffractometer (Advance D8, Bruker, Germany) [40]. The main chemical composition and the XRD patterns of PG are shown in Figure 3. The XRD and XRF results revealed that CaSO_4_-2H_2_O was the main chemical compound in the collected PG, accompanied by small amounts of CaSO_4_-0.5H_2_O and SiO_2_. The chemical composition of PG is shown in Table 1.

The results of the particle size analysis are shown in Figure 4. The results showed that PG particles less than 78.96, 29.34, and 9.9 μm accounted for 90, 50, and 10wt%, respectively, of the total PG particle ranging from 9 to 80 μm, belonging to the ultra-fine particle size of the PGB material.

A 42.5R grade silicate cement with the main components of CaO, SiO_2_, Al_2_O_3_, and Fe_2_O_3_ was selected based on the standard of the silicate cement in China (GB 175-2007). According to the GB/T 17671 test, the 42.5 R grade silicate cement has 3 and 28 d compressive strengths over 22.0 and 42.5 MPa, respectively.

In order to reduce the influence of external impurities on the leaching toxicity test, the standard deionized water was obtained from Shenzhen South China Hi-Tech Water Treatment Equipment Co., Ltd. (Shenzhen, China) and used in this study for the sample preparation and leaching toxicity test.

### 2.3. Experimental Method

The test methods used in this study consisted mainly of the preparation and maintenance of the PG specimens, the leaching toxicity test, and the detection of the main pollutants. The testing process and the main instruments used are shown in Figure 5.

#### 2.3.1. Preparation of the PG Sample

According to the experimental design of PG proportioning, PG was first sieved using a 5 mm standard sieve, then mixed with cement and water using an electric mixer for 10 min according to the requirements of ash-sand ratios of 1:6 (M1), 1:8 (M2), and 1:10 (M3), with 60% wt%. The mold was kept at room temperature for 24 h before demolding [38,39]. The demolded PG specimens were stored in a chamber at a constant temperature and in a humidity chamber (HWS-70B, Teste, Tianjin, China) under laboratory conditions (20 °C and 90% humidity) [41].

#### 2.3.2. Toxicity Leaching Test

According to experience, the PGB body specimens have the highest toxicity leaching when they are maintained for 3 days [14]. Therefore, the solid waste leaching toxicity method (acetate buffer solution method, HJ/T 300-2007) was used to determine the extreme leaching toxicity of the backfilling body conservation for 3 days. First, the PGB body specimen was crushed, then a 100 g sample from the center part of the specimen was ground and sieved using a sieve with an average pore size of 0.045 mm. The sample and leaching agent were mixed in a 2 L polyethylene wide-mouth flask at a solid–liquid ratio of 1:20 and shaken in a fully automatic tilt shaking device (YKC-12) for 18 ± 2 h at a speed and temperature of 30 ± 2 r/min and 23 ± 2 °C. The sample was filtered through a 0.6 μm filter using a Zinten (GM-0.33A) vacuum filter pump after resting the mixture for 2 h. The filtered clear liquor was stored in bottles at 4 °C [40].

#### 2.3.3. Pollutant Indicator Testing

In this study, the inorganic elements and compounds exceeding the detection limit in the PGB were selected according to the leaching toxicity identification standard (GB-5085.3-2007). TP was selected as one of the detection indicators due to the high phosphate contents in PG. In this study, only the TP diffusion in the groundwater was investigated without considering the leaching of other inorganic elements and compounds. The TP content in PG was determined using the ammonium molybdate spectrophotometric method (GB/T 11893-1989).

### 2.4. Mathematical Regression of TP Transport Dispersion Model

After simulating the TP transport in the groundwater in the study area in the steady-state groundwater flow field and predicting its spatiotemporal pattern, a diffusion model of soluble P elements was established to assess the spatiotemporal evolution of TP in the groundwater based on variables related to the source of P pollution and its spatiotemporal distribution. The diffusion transport pattern of TP was regressed for 100 years to provide a basis for future studies. As the model used in this study deals with a high-dimensional problem using three independent variables with a large number of constraints and complex linear relationships [41,42,43], the data were analyzed and solved using the 1stOpt software package (1stOpt 8.0, 7D-Soft High Technology Inc., Beijing, China). The changes were analyzed in the P concentrations at leached TP concentrations of 0.59, 1.88, and 2.46 mg/L, as well as at different times and locations. The obtained results were compared using the determination coefficient (R^2^), the sum of the squared residuals (SSE), and the root mean square error (RMSE) to determine the most effective solved equation in describing the transport law of TP.

## 3. Results

### 3.1. Leaching Toxicity Test

The results of the TP leaching tests were compared with the hazardous waste identification standard-leaching toxicity identification (GB 5085.3-2007). The TP concentrations in leachate with different ash-sand ratios are reported in Table 2.

The results of the leaching toxicity experiments revealed a rapid decrease in the concentration of TP leached from PGB specimens with an increasing cement addition, as shown in Figure 6. When the cement–sand ratio was 1:6 and 1:8, the leaching concentrations of TP were 0.59 and 1.88 mg/L, which were significantly lower than the concentration of 2.46 mg/L when the cement–sand ratio was 1:10. According to the surface water environmental quality standard (GB3838-2002) promulgated by China, in order to meet the standard of centralized drinking water, the detection value of the TP concentration in these areas of rivers, lakes, and groundwater should be less than 0.2 mg/L. Phosphorus in PG exists mainly in the form of phosphate, while ordinary silicate cement mainly consists of CaO, SiO_2_, Fe_2_O_3_, and Al_2_O_3_. During the hydration reaction of cement, soluble phosphate reacts with Ca^2+^, thus adsorbing on the surface of the cement particles and retarding the hydration reaction of the cement, while forming a protective layer of calcium phosphate. Liu studied that in an acidic environment, phosphorus precipitates mainly as H_3_PO_4_, H_2_PO, and CaH_2_PO in a toxic leachate backfilling with PG, while in a neutral environment, phosphate starts to precipitate as small amounts of soluble calcium phosphate (hydroxyapatite and calcium fluorophosphate), and in an environment with an alkaline pH, the increase in Ca^2+^ and F^+^ leads to the precipitation of phosphate in the form of a dilute increased binding of soluble calcium [44,45,46,47,48]. The M1, M2, and M3 groups were considered as the central sources of TP concentrations of the backfilling bodies from which TP is leached to study the diffusion transport of TP in groundwater under different leaching concentrations in the backfilling bodies.

### 3.2. Simulation of Pollutant Transport Using the MT3DMS Model

The main purpose of this study is to interpret the influence of the hydrogeological characteristics of groundwater, atmospheric recharge to the groundwater recharge, and rivers on the TP transport in the study area to provide a basis for the prevention of groundwater pollution in mine areas. Two areas were circled in the eastern and western parts of the study area based on the hydrogeological characteristics to simulate a subsurface backfilling due to the large topographic differences between the eastern and western parts, then the MT3DMS module was used to simulate the distribution and changes in TP concentrations in the study area in the eastern and western parts under different central source concentration conditions.

The results of the MT3DMS module simulation in the western part of the study area are shown in Figure 7. The TP transport pattern in the western part was mainly influenced by topographic features, rivers, and drainage wells. Since there is no significant change in the groundwater head in the western area, the TP transport distance was short, moving towards the eastern part. In addition, the results revealed a convergence in TP concentrations at the drainage wells due to the existence of a well in the northern part, through which groundwater is pumped at a flow rate of 50 m^3^/d. Low TP concentrations can also be diffused toward the southern part as it is transported to the eastern part due to the influence of the drainage ditch and river in the southern part. Although the groundwater head is higher in the eastern and northern parts of the study area and there is an insignificant TP transport toward these directions, a small amount of TP may also exist.

According to the spatiotemporal transport of TP at different concentrations, as shown in Figure 8, there was no exceeding the standard value of 0.2 mg/L in the study area within 500 m for 100 years at the initial source concentrations of 0.59 and 1.88 mg/L. The TP concentrations were less than the standard value of 0.2 mg/L. However, at the initial source concentrations of 2.46 mg/L, TP concentrations of 0.2 mg/L were found within 100, 200, 300, and 400 m at 641, 2358, 5330, and 7959 days, respectively, while the TP concentration within 500 m did not exceed the standard. Therefore, due to the low variation in the groundwater head in the western part of the study area, as well as the TP adsorption and attenuation, exceedances of the TP standard were observed only in the range radius of 400–500 m. In addition, the center of the pollutant concentration moved slightly downstream with an increasing distance.

In the eastern part of the study area, the TP transport was mainly influenced by the changes in the groundwater flow field, as shown in Figure 9. The difference between the groundwater heads in the northern and southern parts of the eastern region was large, while there were two rivers in the south converge, forming a river valley. Therefore, the TP diffusion distance in the eastern part of the study area was long, while that in the eastern and western parts was short. In addition, in the eastern study area, the center TP concentration was obviously shifted gradually to the southern part with the TP transport. TP concentrations may be accumulated northward at the drainage well in the eastern part of the study area, with a flow rate of 300 m^3^/d, during TP in transport to the south, and then be removed from the groundwater through the drainage well.

The spatiotemporal transport and dispersion of TP in the eastern part of the study area at different source concentrations are shown in Figure 10. The results showed lower TP concentrations than the standard value at source concentrations of 0.59 mg/L within 500 m, whereas at the source concentration of 1.88 mg/L, the TP concentration exceeded the standard value at 100 m from the backfilling at 4920 days due to the shift of the source concentration center to the south. No exceedance of the TP concentration was observed in the following 100 years from 200 to 500 m due to the soil sorption as well as the transport attenuation effects. In addition, at a source concentration of 2.46 mg/L, the TP concentration in the central part of the study area exceeded the standard value at 606 days due to the high initial source concentration. This TP concentration exceedance was also observed at 11,517 days within 300 m due to the continuous TP transport. However, no exceedance of the TP concentration was observed within a range of 400–500 m for 100 years. This is due to the large difference between the groundwater head values in the northern and southern parts of the study area, the larger diffusion area, and the attenuation/adsorption effects, causing rapid decreases in TP concentrations.

### 3.3. Spatiotemporal Transport Mechanism of Phosphorus in Groundwater

The simulation results of TP transport in groundwater obtained by the MT3DMS module were solved using the Universal Global Optimization (UGO) algorithm and the 1stOpt software package. The results revealed 119 spatiotemporal transport–diffusion equations for elemental phosphorus in the western region. In this study, only equations with a high relevance to the transport–diffusion in the eastern and western regions were selected, as follows:

Transport–diffusion equation for the western region:(3)$$C_{w}=-0.2334++4.93086C_{0}T-2.09696DT+0.04863C_{0}+0.0003D-0.00031C_{0}{}^{2}-9.43824D^{2}$$

Transport–diffusion equation for the eastern region:(4)$$C_{e}=-5.48\times 10^{-3}-3.78\times 10^{-5}C_{0}T-5.54\times 10^{-10}C_{0}D+0.032C_{0}+7.34\times 10^{-5}D-3.61\times 10^{-8}D^{-0.01096}-3.61\times 10^{-8}D^{2}$$
where *C_w_* and C*_e_* denote the instantaneous concentration of TP at a certain time and in a certain area (mg/L) in the western and eastern parts of the study area, respectively; *C*_0_ denotes the source concentration (mg/L); *T* is the time (day); and *D* is the distance (m). The fitted eigenvalues of the equations for different regions are shown in Table 3.

The coefficients of determination of the diffusion equations of the pollutants in the east and west were 0.73 and 0.85, respectively, which were closer to 1. The independent variables of the source concentration, time, and distance explained the dependent variable of the diffusion concentration to a high degree, which had some reference value for future studies of the TP concentration in different regions at different times. Meanwhile, according to the RMSE values of the fitted equations, we can see that both fitted equations have some errors, so the fitted equations only provide some reference basis for the diffuse transport values of TP.

## 4. Conclusions

The aim of this study was to investigate the spatiotemporal transport of TP from PGB bodies in the groundwater environment of a mine in Anhui Province and to reveal the transport pattern and diffusion mechanisms of TP in groundwater using numerical groundwater simulations combined with leaching toxicity experiments.

(1) Through leaching toxicity experiments, we found that the leaching concentration of TP was different for the cement mixing quantity, and the TP leaching concentration was negatively correlated with the cement blending. Meanwhile, Ca^2+^ in cement reacts with PO_4_^3^ in the cement hydration reaction and solidifies PO_4_^3−^, thus reducing the leaching of TP.

(2) Through the simulation of both the eastern and western regions, TP pollution does not occur within 100 m in 100 years. However, with the increase in the source concentration, TP in the process of transport, the concentration exceeds the standard in a certain spatial and temporal area. Moreover, the eastern part has a larger topographic difference than the western part, and the TP transports farther and the concentration center will be shifted to the south, but the TP concentration also changes faster and has a larger dispersion area due to the topography.

(3) The coefficients of determination of the fitted equations for the east and west were 0.75 and 0.85. Both the spatial and temporal variation and the source concentration had an effect on the transport of TP. The coefficient of determination of the fitted equation in the east is lower because of the large variation in the head in the east. In contrast, the western part is relatively flat and the TP pollution diffusion is less variable, so the TP determination coefficient is better.

## Figures and Tables

**Figure 1 ijerph-19-14957-f001:**
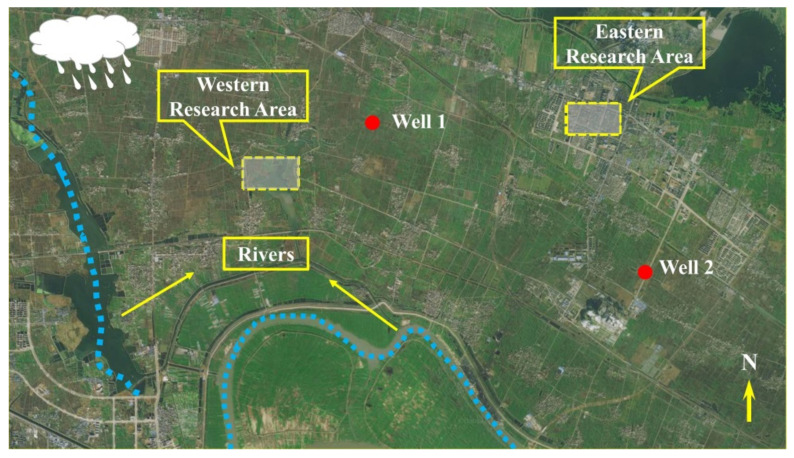
Determination of the decay rate constant according to Guggenheim’s method.

**Figure 2 ijerph-19-14957-f002:**
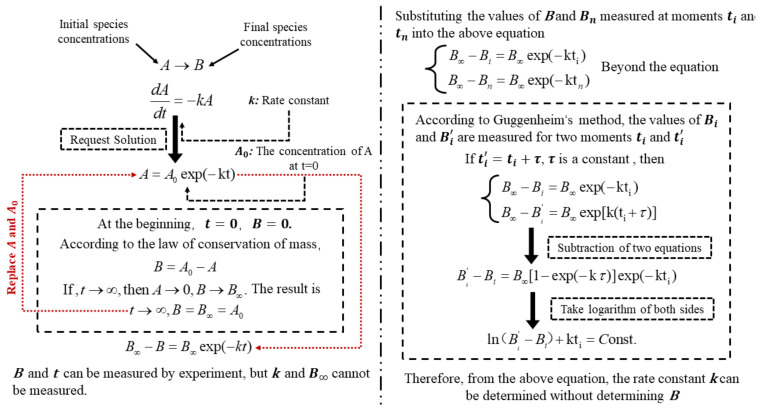
Determine the decay rate constant according to Guggenheim’s method.

**Figure 3 ijerph-19-14957-f003:**
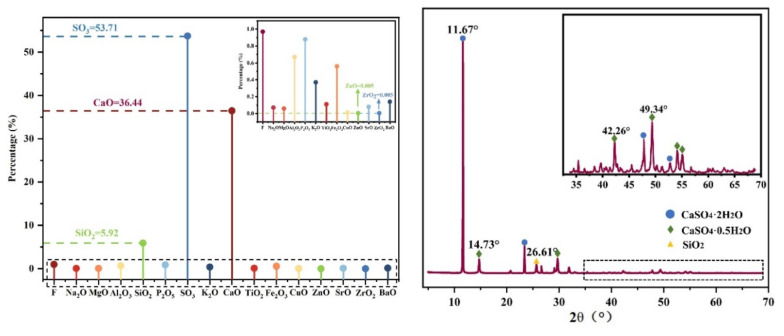
XRD and XRF patterns of phosphogympsum.

**Figure 4 ijerph-19-14957-f004:**
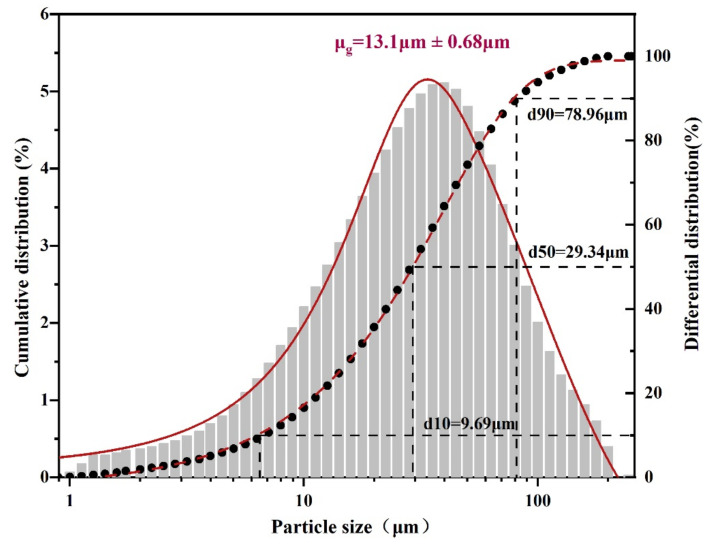
Results of PG particle size analysis.

**Figure 5 ijerph-19-14957-f005:**
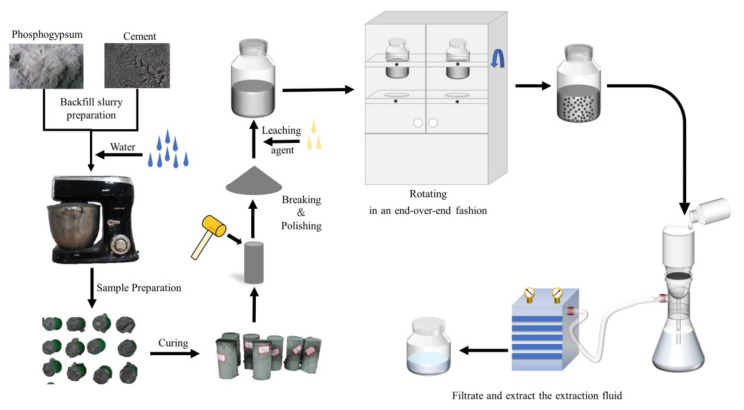
Flowchart of the experimental approach used in this study.

**Figure 6 ijerph-19-14957-f006:**
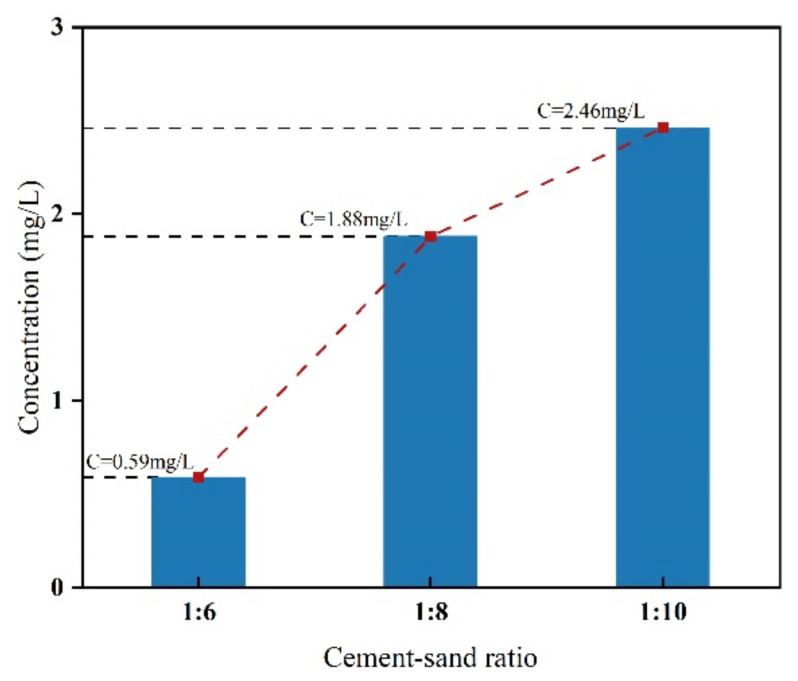
Toxicity leaching results.

**Figure 7 ijerph-19-14957-f007:**
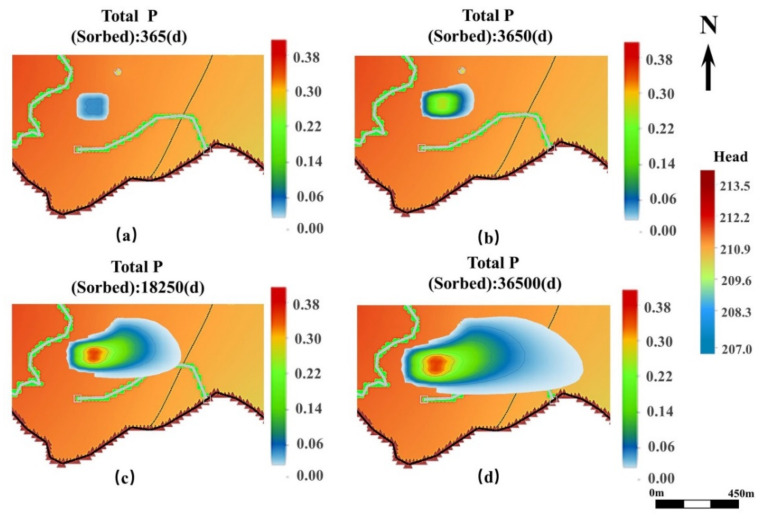
MT3DMS simulation results of the horizontal TP transport in the western part of the study area. (**a**–**d**) shows the transport of TP after 1 year, 10 years, 50 years, and 100 years of groundwater dispersion, respectively.

**Figure 8 ijerph-19-14957-f008:**
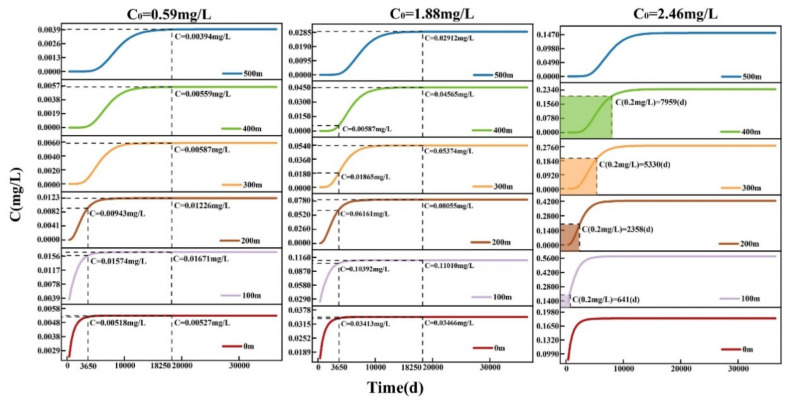
Changes in the spatiotemporal transport of TP in the western part of study area under different source concentrations.

**Figure 9 ijerph-19-14957-f009:**
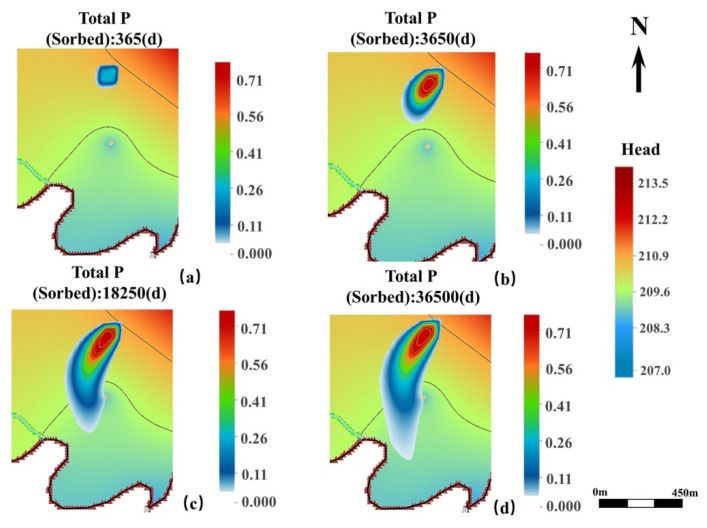
MT3DMS simulation results of the horizontal TP transport in the eastern part of the study area. (**a**–**d**) shows the transport of TP after 1 year, 10 years, 50 years, and 100 years of groundwater dispersion, respectively.

**Figure 10 ijerph-19-14957-f010:**
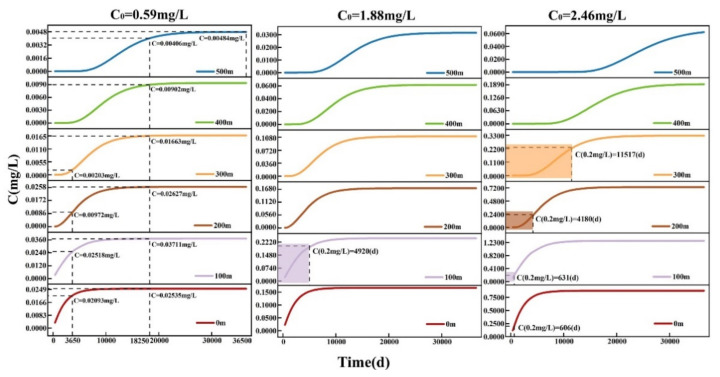
Changes in the spatiotemporal TP transport in the eastern part of the study area under different source concentrations.

**Table 1 ijerph-19-14957-t001:** Chemical composition of phosphogympsum.

Chemical Composition	F	Na_2_O	MgO	Al_2_O_3_	SiO_2_	P_2_O_5_	SO_3_	K_2_O
Mass fraction/%	0.97	0.069	0.056	0.662	5.92	0.875	53.72	0.368
Chemical composition	CaO	TiO_2_	Fe_2_O_3_	CuO	ZnO	SrO	ZrO_2_	BaO
Mass fraction/%	36.44	0.106	0.573	0.013	0.003	0.076	0.006	0.14

**Table 2 ijerph-19-14957-t002:** TP concentrations in leachate from PGB bodies under different cement-sand ratio.

Number	M1	M2	M3
Cement-sand ratio	1:6	1:8	1:10
TP concentration (mg/L)	0.59	1.88	2.46

**Table 3 ijerph-19-14957-t003:** Fitting eigenvalues for different regional equations.

Region	Eigenvalue		
	RMSE	R^2^	SSE
Western	0.13	0.85	29.87
Eastern	0.08	0.73	12.57

## Data Availability

Not applicable.

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
