# Peer review of "The Phosphorus Transport in Groundwater from Phosphogypsum-Based Cemented Paste Backfill in a Phosphate Mine: A Numerical Study"

_ijerph, 2022, doi:10.3390/ijerph192214957_

Round 1

Reviewer 1 Report

This paper investigates the spatiotemporal transport of TP from PGB in the groundwater environment using numerical groundwater simulations and leaching toxicity experiments. The study is important for green mining and ecological civilization. In my opinion, the manuscript could be published in International journal of Environmental Research and Public Health after minor revision. Followings are comments and suggestions:

1 The first sentence of the second paragraph of the introduction should be expressed as "PG backfill treatment" instead of "PG discharge".

2 Abbreviations appearing for the first time should be in full, e.g., PGB, as sated in line 47 of the introduction.

3 In line 236, how the conclusion was obtained, please give reasons and justify through literature.

4 The illustration in Figure 3 does not match the picture, please revise it and adjust the format of the figure and table in the context.

5 The correct form of compressive strength unit should be "MPa", please correct it.

6 In Section 3.2, the simulation results of Figure 7 and Figure 9 are briefly explained for better understanding the principles, also unify the time units mentioned in the whole section.

7 In Section 3.2, please cite some literature to provide a theoretical basis for the analysis of the results.

8 The manuscript also should be properly proofread to eliminate all typos, please double check the grammar, and improve the English level throughout the manuscript. e.g.,

9 In line 203, what does it mean where "θ" appears.

10 In line 237, the term "leaching" appears twice.

11 There are some new biomethods for treating phosphogypsum. Please cite it in the introduction:

[1] Xiang J, Qiu J, Zheng P, Sun X, Zhao Y, Gu X. Usage of biowashing to remove impurities and heavy metals in raw phosphogypsum and calcined phosphogypsum for cement paste preparation. Chemical Engineering Journal 2023; 451: 138594.

Reviewer 2 Report

In this manuscript, the authors conducted considerable tests and numerical simulation to demonstrate the migration property of total phosphorus from phosphogypsum-backfill to groundwater environment. The findings are very significant to improve the protection of groundwater environment during mining backfill. Overall, minor revision is suggested for following comments should be addressed before publication.

1.       Please refine the abstract, the sentence of which is not concise and accurate enough.

2.       Section 2.3.2, there is no reference support.

3.       Section 2.1, how to determine groundwater head height?

4.       There are many abbreviations in this article. The authors are suggested to illustrate the abbreviations uniformly.

5.       The format should be improved, e.g., size of the formula should be uniform

6.       In the results section, there are two units of years and days in the analysis process, but the year does not appear in the figure and it is recommended to unify the units.

7.       Line 203. What does θ mean ?

8.       The Figure 8 and 10 are not clear enough and need to be adjusted.
